# Multifaceted Functions of Host Cell Caveolae/Caveolin-1 in Virus Infections

**DOI:** 10.3390/v12050487

**Published:** 2020-04-26

**Authors:** Yifan Xing, Zeyu Wen, Wei Gao, Zhekai Lin, Jin Zhong, Yaming Jiu

**Affiliations:** 1The Center for Microbes, Development and Health, Key Laboratory of Molecular Virology and Immunology, Institut Pasteur of Shanghai, Chinese Academy of Sciences, Shanghai 200031, China; 2University of Chinese Academy of Sciences, Yuquan Road No. 19(A), Shijingshan District, Beijing 100049, China; 3Key Laboratory of Molecular Virology and Immunology, Institut Pasteur of Shanghai, Chinese Academy of Sciences, Shanghai 200031, China

**Keywords:** caveolae, caveolin-1, virus life cycle, virus entry, virus trafficking, virus replication, virus assembly, virus egress, signaling pathway

## Abstract

Virus infection has drawn extensive attention since it causes serious or even deadly diseases, consequently inducing a series of social and public health problems. Caveolin-1 is the most important structural protein of caveolae, a membrane invagination widely known for its role in endocytosis and subsequent cytoplasmic transportation. Caveolae/caveolin-1 is tightly associated with a wide range of biological processes, including cholesterol homeostasis, cell mechano-sensing, tumorigenesis, and signal transduction. Intriguingly, the versatile roles of caveolae/caveolin-1 in virus infections have increasingly been appreciated. Over the past few decades, more and more viruses have been identified to invade host cells via caveolae-mediated endocytosis, although other known pathways have been explored. The subsequent post-entry events, including trafficking, replication, assembly, and egress of a large number of viruses, are caveolae/caveolin-1-dependent. Deprivation of caveolae/caveolin-1 by drug application or gene editing leads to abnormalities in viral uptake, viral protein expression, or virion release, whereas the underlying mechanisms remain elusive and must be explored holistically to provide potential novel antiviral targets and strategies. This review recapitulates our current knowledge on how caveolae/caveolin-1 functions in every step of the viral infection cycle and various relevant signaling pathways, hoping to provide a new perspective for future viral cell biology research.

## 1. Introduction

Since 2009, the World Health Organization (WHO) has announced a total of six “public health emergencies of international concern”, all of which are related to virus infections [1], indicating the huge threat of virus infections to humankind. Viruses, as complete parasites, accomplish their infection by hijacking various host mechanisms; for example, ribosomes of host cells are essential for all viruses to synthesize their own proteins [2], the host cytoskeleton system and molecular motors dramatically reorganize to facilitate key infection processes [3], a large number of viruses utilize clathrin-mediated endocytosis to complete invasion [4], and so on. Among those host antiviral mechanisms and hijacking machinery, there is a very important, yet less explored one, the caveolin-1 and formed membrane invagination caveolae-mediated virus infection.

Mechano-sensing caveolae are considered to be a specialized form of lipid raft, which shows flask-shaped membrane invagination with a diameter of 50–100 nm enriched in cholesterol, phospholipids, and sphingolipids [5,6]. The basic components of caveolae are caveolins (caveolin-1, -2, -3), cavins (cavin-1, -2, -3), Eps15-homology domain-containing protein 2 (EHD2), protein kinase C and casein kinase substrate in neurons 2 (PACSIN2), and the receptor tyrosine kinase-like orphan receptor 1 (ROR1) [5]. Among these, caveolin-1 (CAV-1) is the principal structural protein involved in multiple critical biological processes, including cholesterol homeostasis, cell mechano-sensing, and signal transduction. A number of human diseases are related to mutations or loss of CAV-1, such as lipodystrophy, arterial hypertension, and cancers [5]. Importantly, accumulating studies revealed the versatile and multithreaded functions of caveolae and CAV-1 during virus infection [4,7,8]. 

From the host cell perspective, viruses undergo entry, intracellular trafficking, replication, assembly, and egress steps to complete their infection, which is referred to as the viral life cycle. Previous works have indicated that caveolae and CAV-1 play significant roles in every step of the viral life cycle with the focus on entry, and consequently is essential for productive infection [7]. Simian virus 40 (SV40) has been widely studied and proposed to enter cells through caveolae-mediated endocytosis [9,10,11,12,13,14], and the subsequent intracellular trafficking also requires CAV-1 to go through “a two-step transport route” [12]. Studies on human immunodeficiency virus (HIV) revealed multiple ways in which CAV-1 participates in signaling regulation during infection [8]. Researchers have started to have deeper insights into the important mechanistic associations of caveolae and CAV-1, with all the stages during virus infection, despite that our understanding of the responses of infection-triggered cellular caveolae system is far from complete.

In this review, the multifunctional roles of caveolae, particularly CAV-1, in different infection stages, as well as related signaling pathways, will be comprehensively summarized and discussed. We have compiled all the caveolae relevant viruses, which will sketch a “caveolae/CAV-1-virus crosstalk” blueprint and give a new sight for antiviral strategies.

## 2. The Role of Caveolae/CAV-1 in Virus Entry

The initial step of the viral life cycle is binding to the cell surface and entering host cells through recognition with specific receptors. Viruses enter host cells via several endocytic pathways: clathrin-mediated uptake, which is widely utilized by numerous viruses; caveolae/cholesterol-dependent endocytosis, which can be used as a substitute to or also supplement the clathrin route; macropinocytosis [4]. The caveolae pathway occurs not only in enveloped viruses but also in non-enveloped viruses. Here, we introduce the regulatory function and molecular mechanisms of caveolae dependent virus entry in four categories based on human/non-human hosts and enveloped/non-enveloped viruses. Additionally, we compile examples of different viruses infecting specific cells using the caveolae-mediated pathway in Table 1.

### 2.1. The Role of Caveolae/CAV-1 in the Entry of Enveloped Human Viruses

Japanese encephalitis virus (JEV) is an enveloped, zoonotic mosquito-borne flavivirus, causing serious neurological diseases in Asia [15]. A previous study found that clathrin is not required for JEV entry into B104 rat neuroblastoma cells. Inactivation of CAV-1 with small interfering RNA (siRNA) or dominant-negative mutants decreases the endocytosis of JEV [16]. Moreover, drug treatment showed that the entry of JEV is dynamin- (GTPase function as scissors to separate vesicles) and acidic pH-dependent [17]. Further research revealed that CAV-1-mediated endocytosis is indispensable for JEV entry into another cell line, human neuroblastoma SK-N-SH cells [18]. First, viral envelope protein binds to the cell receptor and then activates the EGFR-PI3K-RhoA-ROCK-CFL1 signaling pathway, leading to F-actin polymerization and CAV-1 phosphorylation. Subsequently, the phosphorylation of CAV-1 induces Rac1 activation, initiates PAK1-CFL1-mediated actin polymerization, and facilitates virus internalization via dynamin-dependent caveolae-mediated endocytosis. It is worth noting that the entry of JEV into BHK-21, HeLa, and PK15 cells is clathrin-mediated [19,20,21], suggesting that a virus may use a distinct mechanism to enter different cells.

Hepatitis B virus (HBV) is an encapsulated DNA virus belonging to the *Hepadnaviridae* family, and its persistent infection may lead to hepatocellular carcinoma (HCC) [22]. The internalization of hepatitis B surface antigen (HBsAg) is blocked when cells are pre-treated with methyl-β-cyclodextrin (MβCD), a drug that does not affect clathrin-mediated uptake but impairs caveolae-dependent endocytosis via cholesterol depletion [23]. Another study has shown that HBV infection in HepaRG cells is inhibited by 35% when the cholesterol level is reduced with nystatin (Ny) and MβCD treatment, but not with NH4Cl and bafilomycin A1 (Baf) which effectively block the vacuolar H^+^ ATPase pumps [24]. Taking these two studies together, the uptake of HBV to host cells is through the caveolae-meditated, acidic pH-independent way. It is worth to note that the different pH requirements for JEV and HBV infections may be due to their different intracellular transport routes, which will be discussed in the “trafficking” section.

**Table 1 viruses-12-00487-t001:** Summary of viruses that enter host cells via caveolae-mediated pathways.

Classification	Viruses	Family	Host Cells	Experimental Approaches	References
Human enveloped viruses	JEV	*Flaviviridae*	Rat neuroblastoma cells, human neuroblastoma SK-N-SH cells	Chemical inhibitors, RNAi, dominant-negative constructs, Fluorescence imaging	[16,18]
	HBV	*Hepadnaviridae*	COS-7 cells, HepaRG cells	Live cell imaging, chemical inhibitors, dominant-negative constructs	[23,24]
	HCoV-229E	*Coronaviridae*	Human fibroblasts L132	Fluorescence imaging, electron microscopy, siRNA, chemical inhibitors	[25]
	HCoV-OC43	*Coronaviridae*	HCT-8 cells	Fluorescence imaging, siRNA, chemical inhibitors	[26]
	RSV	*Paramyxoviridae*	Cattle dendritic cells	Chemical inhibitors, fluorescence imaging	[27]
	Filoviruses (EBOV, MARV pseudotype viruses)	*Filoviridae*	293T cells and Hela cells	Chemical inhibitors, fluorescence imaging, internalization kinetics	[28]
	RVFV	*Phenuiviridae*	HeLa, HepG2, and 293T cells	Chemical inhibitors, RNAi, dominant-negative constructs, fluorescence imaging	[29]
Animal enveloped viruses	EHV-1	*Herpesviridae*	Equine brain microvascular endothelial cells	Chemical inhibitors, dominant-negative constructs, fluorescence imaging	[30]
	CSFV	*Flaviviridae*	Macrophages	Digital gene expression profiling, fluorescence imaging, siRNA, CAV-1 overexpression	[31]
	TGEV	*Coronaviridae*	Swine testis cells	Live cell imaging, chemical inhibitors	[32]
	PEDV	*Coronaviridae*	Vero cells and IPEC-J2 cells	Internalization kinetics, chemical inhibitors, fluorescence imaging, fractionation	[33]
	TFV	*Iridoviridae*	HepG2 cells	Chemical inhibitors, CAV-1 peptide, fluorescence imaging	[34]
	ISKNV	*Iridoviridae*	MFF-1 cells	Internalization kinetics, chemical inhibitors, fluorescence imaging, far-western blotting, pulldown, coimmunoprecipitation, siRNA, truncation, fractionation	[35,36,37]
	IBV	*Coronaviridae*	Vero cells	Fractionation, fluorescence imaging, chemical inhibitors	[38]
	CRCoV	*Coronaviridae*	HRT-18G cells	Chemical inhibitors, fluorescence imaging, siRNA	[39]
	PPRV	*Paramyxoviridae*	Caprine endometrial epithelial cells	Internalization kinetics, chemical inhibitors, fluorescence imaging, siRNA, electron microscopy	[40]
	A-MLV	*Retroviridae*	NIH 3T3 cells	Internalization kinetics, chemical inhibitors, immunoprecipitation, fluorescence imaging	[41]
Human non-enveloped viruses	SV40	*Polyomaviridae*	CV-1 cells	Internalization kinetics, chemical inhibitors, fractionation	[9]
	HAdV-D	*Adenoviridae*	Corneal cells	Chemical inhibitors, fluorescence imaging, siRNA, fractionation, electron microscopy, animal model	[42]
	HAdV-C	*Adenoviridae*	Plasmocytic cell lines	Chemical inhibitors, dominant-negative constructs, fluorescence imaging	[43]
	BKV	*Polyomaviridae*	Vero cells and human renal proximal tubular epithelial cells	Internalization kinetics, chemical inhibitors, dominant-negative constructs	[44,45]
	HPV-31	*Papillomaviridae*	COS-7 cells and human keratinocytes	Electron microscopy, chemical inhibitors, dominant-negative constructs, internalization kinetics, fluorescence imaging, flotation	[46,47,48]
	EV1	*Picornaviridae*	SAOS cells	Fluorescence imaging, electron microscopy, dominant-negative constructs, fractionation	[49]
	EV71	*Picornaviridae*	Jurkat T and mouse L929 cells	siRNA, chemical inhibitors, fluorescence imaging	[50]
	Reovirus	*Reoviridae*	A549, Hela and HEK-293 cells	Chemical inhibitors, dominant-negative constructs, fluorescence imaging	[51]
Animal non-enveloped viruses	FMDV	*Picornaviridae*	MCF-10A cells	Fluorescence imaging, chemical inhibitors, siRNA	[52]
	MPyV	*Polyomaviridae*	NIH 3T6 fibroblasts and NMuMG epithelial cells	Electron microscopy, chemical inhibitors, fluorescence imaging	[53]
	ARV	*Reoviridae*	Vero and DF-1 cells	Chemical inhibitors, siRNA, coimmunoprecipitation, fluorescence imaging	[54]
	MDRV	*Reoviridae*	Vero and DF-1 cells	Chemical inhibitors, fluorescence imaging	[55]
	GCRV	*Reoviridae*	CIK cells	Chemical inhibitors, fluorescence imaging, dominant-negative constructs	[56]

Coronaviruses are a large class of RNA viruses with an envelope and a linear single positive strand in their genome and they exist widely in nature. There are currently seven known coronaviruses that can infect humans, two of which are related to caveolae, namely, human coronavirus 229E (HCoV-229E) and human coronavirus OC43 (HCoV-OC43). The aggregated CD13, a receptor for HCoV-229E, is identified to be co-localized with CAV-1 in human fibroblasts. The binding of HCoV-229E to CD13 triggers the cross-linking of this receptor, and thereby HCoV-229E reaches toward the caveolae microdomain for entry [25]. HCoV-OC43 employs CAV-1 for internalization in HCT-8 cells, and the subsequent trafficking is dynamin- and actin-dependent [26]. Besides, whether CAV-1 plays a role in the entry of severe acute respiratory syndrome coronavirus (SARS-CoV) remains controversial. Bioinformatics studies show that CAV-1 and lipid raft is related to SARS-CoV infection, as in silico analysis presents tens of CAV-1 binding domains (CBD) in SARS-CoV proteins [57]. CBD has been defined to bind CAV-1 via the caveolin scaffolding domain (CSD) by phage display techniques and a random peptide library [58]. However, existing experimental evidence indicates that SARS-CoV enters in a clathrin-mediated manner [59,60]. Why does there exist an abundance of CBDs in SARS-CoV proteins when SARS-CoV seemingly prefers clathrin-mediated-endocytosis? Byrne et al., using bioinformatics tools, examined the role of CBD in CAV-1 interactions. Evidence shows that the key amino acid residues of CBD are folded inside the protein, thus may not directly interact with CAV-1. They concluded that the interfaces between CAV-1 and targets may be more structurally diverse than presently appreciated, reminding us that CBD is not a sufficient condition for interaction [61]. This may partly explain why SARS-CoV undergoes a clathrin-mediated manner but not the caveolae pathway. 

Respiratory syncytial virus (RSV) is an enveloped RNA virus belonging to the family of *Paramyxoviridae* and is the most common cause of pediatric viral pneumonia. A previous study showed that RSV uptake in cattle dendritic cells is sensitive to phorbol myristate acetate (PMA) and filipin, which can specifically inhibit the virus entering cells through caveolae. Furthermore, caveolae and RSV antigen are observed to co-localize by confocal microscopy. Thus, this study pointed out that the uptake of RSV in cattle dendritic cells is mediated by caveolae, revealing a new way by which antigen uptakes in dendritic cells [27]. 

Filovirus, such as Ebola virus (EBOV) and Marburg virus (MARV), are highly pathogenic to humans. Evidence showed that both Zaire EBOV pseudotype viruses and MARV pseudotype viruses utilize caveolae endocytosis as their way to enter cells [28]. Notably, caveolae-mediated endocytosis can only be shown with pseudotype viruses but not with real EBOV, which requires macropinocytosis [62]. Rift valley fever virus (RVFV), a *Phenuiviridae* family virus, transmits to humans mainly through mosquito bites or contact with infected animals and causes rift valley fever (RVF). Through dominant-negative protein expression and RNA interference (RNAi), Harmon et al. showed that RVFV MP-12, an attenuated strain of RVFV, enters multiply cell lines through the dynamin-dependent, CAV-1-mediated pathway [29].

### 2.2. The Role of Caveolae/CAV-1 in the Entry of Enveloped Animal Viruses

Enveloped animal viruses are reported to be associated with caveolae as well. Equine herpes virus (EHV) is a widespread DNA virus, causing respiratory diseases, abortion, and neurologic diseases in horses [63]. EHV-1 co-localizes with CAV-1 in equine brain microvascular endothelial cells (EBMECs), and there is a great decrease of infectivity when expressing the dominant-negative form of equine CAV-1. In addition, EHV-1 infection could cause CAV-1 phosphorylation and may be related to subsequent signal transduction [30].

Classical swine fever (CSF), one of the major infectious diseases threatening the pig industry, is caused by the infection of the classical swine fever virus (CSFV). In CSFV Shimen-infected macrophages, the transcription and translation levels of CAV-1 are significantly upregulated. In addition, the expression of CSFV Shimen E2 protein and CAV-1 appear to be temporally well-correlated in infected cells. Meanwhile, the cell membrane CAV-1 is significantly enhanced during the course of infection, suggesting that the invasion of CSFV Shimen into macrophages via a CAV-1-mediated pathway. Notably, the caveolae pathway does not seem to be the only way for CSFV Shimen to enter macrophages since CAV-1 siRNA treatment did not prevent viral replication entirely [31]. Transmissible gastroenteritis virus (TGEV), another pig-hosted virus, can enter cells through the clathrin-mediate pathway. Nevertheless, it was subsequently discovered that the entry of TGEV is not completely prevented when clathrin-mediated endocytosis and the caveolae-dependent route are abolished. This study visualized the TGEV internalization by single-virus tracking for the first time, and both clathrin- and caveolae-mediated routes were employed by TGEV in the same host cell by using clathrin or caveolae inhibitors [32]. Porcine epidemic diarrhea virus (PEDV), causing contact intestinal infectious disease, also utilizes both clathrin- and caveolae-mediated routes to enter into both Vero and IPEC-J2 cells. Moreover, the endocytosis efficiency of different PEDV subtypes is dramatically changed, which reminds us that comparing the differences between subtypes may reveal key information that determines the virus invasion mechanism [33].

Tiger frog virus (TFV), an *Iridoviridae* family member, is the main pathogen that causes the death of farmed tiger frog tadpoles. TFV enters HepG2 cells via caveolae-mediated endocytosis and subsequently transports into the Golgi complex as caveola-internalized cargo [34]. The infection spleen and kidney necrosis virus (ISKNV) also belongs to the *Iridoviridae* family, causing spleen and kidney necrosis and resulting in explosive deaths in freshwater aquaculture. Similar to TFV, its invasion to cells is reported via caveolae-mediated endocytosis as well [35,36,37].

Avian infectious bronchitis virus (IBV), an enveloped RNA virus, is a member of the *Coronaviridae* family. The structural proteins of IBV are co-localized with CAV-1, and MβCD or mevastatin treatment largely impairs the viral cycle by preventing virus particles from binding to the cell surface [38]. Canine respiratory coronavirus (CRCoV; causes an acute intestinal infectious disease characterized by vomiting, diarrhea, dehydration, and recurrence), which is also a member of *Coronaviridae* family, has been reported to hijack caveolae endocytosis to enter HRT-18G cells by using chemical inhibitors of pathways involved in viral entry, siRNA, and confocal microscopy assays [39]. Besides, through methods similar to the above cases (summarized in Table 1), the enter of peste des petits ruminants virus (PPRV; mainly infects small ruminants such as goats, sheep, and spreads in west Africa and central parts of Asia) into caprine endometrial epithelial cells and amphotropic murine leukemia virus (A-MLV; an RNA tumor virus causing leukemia in mice) into NIH 3T3 cells are also proven to be caveolae-dependent [40,41].

### 2.3. The Role of Caveolae/CAV-1 in the Entry of Non-Enveloped Human Viruses

In addition to enveloped viruses, some non-enveloped viruses also utilize caveolae-mediated endocytosis to enter the host cells. SV40, a non-enveloped virus infecting both humans and monkeys, is the first virus reported to enter cells via the caveolae pathway. Once bonded to major histocompatibility complex (MHC) class I antigens on the cell surface, SV40 induces a transient breakdown of actin stress fibers and thus forms numbers of actin tails, which results from SV40-induced local tyrosine phosphorylation of caveolae [9,10,11,12,13,14].

Human adenoviruses (HAdVs), a non-enveloped double-stranded DNA virus, utilize clathrin-mediated endocytosis to enter cells in most cases [64,65]. However, the entry of HAdV-D to corneal cells requires lipid raft microdomains and CAV-1, and immunoelectron microscopy confirms the presence of HAdV-D37 in caveolae-like structures [42]. Another HAdV subtype, HAdV-C, has also been shown to enter plasmacytic cells via lipid raft/caveolae endocytosis by using chemical inhibitors, dominant-negative constructs, and imaging methods [43]. These two studies revealed that HAdVs are able to take advantage of all possible pathways to ensure their entry into different host cells.

Human polyomavirus BK (BKV) is another non-enveloped virus that causes severe polyomavirus-associated nephropathy. The uptake of BKV to both Vero cells and human renal proximal tubular epithelial cells (HRPTEC) is via a CAV-1-mediated, cholesterol-needed, and clathrin-independent endocytosis [44,45]. By treating tyrosine kinase inhibitor genistein, the life cycle of BKV is strongly inhibited in the initial step of infection, indicating the essential role of the caveolae-associated signaling pathway in the BKV productive infection [44]. The entry of human papillomavirus (HPV) 31 to COS-7 cells and human keratinocytes is through the caveolae pathway, and the virions are transiently co-localized to the caveosome once they enter into the cells [46,47,48].

The host cell receptor α2β1 integrin of *Picornaviridae* Echovirus 1 (EV1) is accompanied by CAV-1, and expression of a dominant-negative CAV-1 inhibits EV1 infection [49]. In alignment with EV1, multiple lines of evidence suggest that initial infection of human PSGL-1-mediated Enterovirus 71 (EV71) is through the caveolae-mediated pathway [50]. Moreover, CAV-1 and membrane cholesterol are also found to be indispensable in the early step of reovirus virions infection [51].

### 2.4. The Role of Caveolae/CAV-1 in the Entry of Non-Enveloped Animal Viruses

Foot-and-mouth disease virus (FMDV) is a picornavirus that utilizes integrins and heparan sulfate (HS) as host cell receptors [66]. Studies have provided evidence that HS-mediated entry of FMDV is caveolae-associated [52], while integrins-mediated viral entry is clathrin-dependent [67]. The entry mechanisms of FMDV showed that the preferences for the viral internalization way are associated with specific receptors which they bind to. Electron microscopy revealed that mouse polyomavirus (MPyV) or its artificial virus-like particles enter cells through vesicles delivered from the caveolar microdomain. In line with this, MβCD treatment causes a considerable reduction of infection in NIH 3T6 fibroblasts and NMuMG epithelial cells [53], suggesting that cholesterol-rich lipid raft microdomains are essential for MPyV infection.

Viruses in the *Reoviridae* family, including Avian reovirus (ARV), Muscovy duck reovirus (MDRV), and Grass carp reovirus (GCRV), use caveolae for internalization. ARV infection activates p38 mitogen-activated protein kinase (MAPK) and protein tyrosine kinase Src and induces the phosphorylation of CAV-1 (Tyr14) and Rac1 activation, which help its entry [54]. MDRV enters DF-1 and Vero cells via the caveolae route, and dynamin and acidic endosomal environment are essential for its successful infection [55]. Analogously, the entry of GCRV also requires caveolae-mediated endocytosis [56].

In this section, we thoroughly discussed the human and animal, enveloped and non-enveloped viruses which take caveolae/CAV-1 as an entry mechanism (summarized in Table 1). This knowledge was mainly gained by using specific drugs (summarized in Figure 1), RNAi, dominant-negative mutants targeting different endocytosis pathways, as well as imaging methods, and we have summarized the specific methods of each virus in Table 1. We also draw a pattern diagram about the entry mechanism of the viruses mentioned in this section in Figure 1. Despite all of these extensive studies, no simple rule has been discovered regarding what determines the usage of endocytosis pathways by a given virus. We need to pay attention that certain specific viruses may utilize different endocytosis routes when entering different cells. Besides, inhibition of clathrin- and caveolae-mediate endocytosis both affect infection in some cases, indicating the possibility of coexistence between the two pathways. Unfortunately, we did not find a pattern of which endocytosis the virus chooses, at least in terms of host cells. Moreover, there might be a complex crosstalk between endocytic pathways. We found that some family of viruses, such as coronaviruses and iridovirus, are more likely to use the caveolar pathway to enter target cells, but the existence of common machinery or mechanism needs further exploration. The utilization of caveolae is also related to the subsequent intracellular process of viruses, but we cannot draw a clear picture yet. The reasons why viruses choose the caveolae-mediated pathway and mechanisms of how to achieve entry by utilizing this pathway still require extensive study.

## 3. The Role of Caveolae/CAV-1 in Virus Intracellular Trafficking

Cargo entering via caveolae-mediated endocytosis normally moves to CAV-1 enrichment caveosomes. The vesicles are then transported to Golgi or ER, which bypass the acidic endosome system [68]. The trafficking of clathrin-mediated endosomes relies on acidic pH, while the transport of CAV-1 containing vesicles to the destination is neutral pH select, which is one way to distinguish these two ways. Nevertheless, there is crosstalk between multiple viral intracellular trafficking pathways, which might share partially overlapped mechanisms, but their dependence on CAV-1 is ambiguous.

The presence of a two-step transport route of SV40 is a classic caveolae-associated trafficking pathway. Firstly, SV40 enter caveosomes after detached from the plasma membrane with caveolar vesicles, and then the viruses are sorted into the smooth ER compartment via tubular, protein-free membrane vesicles, which move rapidly along microtubules [12]. However, SV40 could also use the CAV-1 independent pathway. In CAV-1 deficient cells, SV40 viruses pass through caveosome-like organelles lacking CAV-1 and transfer to ER [69]. ISKNV, which takes the caveolae route for its entry, is suggested to go through the two-step transport route similar to SV40 [35]. These two cases raise a potential correlation between caveolae/CAV-1 mediated entry and the subsequent transportation within target cells.

The transportation of MPyV, which takes caveolae-mediated endocytosis for entry, to the perinuclear region is through the CAV-1-containing monopinocytic vesicles and the involvement of the cytoskeletal system. Moreover, the co-localization of polyomavirus VP1 and CAV-1 is throughout the whole trafficking process, and the frequent fusion of empty vesicles derived from caveolae with virus-containing vesicles is observed [53]. Furthermore, the chimeric adenoviral vectors HAdV2/BAdV4 are able to induce the formation of mega-caveosomes after internalization within caveolae microdomain, suggesting that the subsequent trafficking does not require the involvement of endosomes [70].

Caveolae- and clathrin-mediated endocytic pathways and the subsequent transportation were once thought to be independent, but now there is increasing evidence that several viruses can exploit crosstalk between these routes. Human papillomavirus type 16 (HPV16), JC virus (JCV), and bovine papillomavirus type 1 (BPV1) have been determined to enter through clathrin-mediated endocytosis, but the subsequent route is CAV-1-dependent [71,72,73]. On the contrary, FMDV is suggested to enter cells via caveolae-mediated endocytosis and then trafficks with endosomes in MCF-10A cells [52]. Moreover, a similar caveolae-to-endosome transport route has also been identified in HPV31 infection that is Rab5 GTPase-dependent [48]. Additionally, a recent study revealed that PEDV enters cells through both clathrin- and caveolae-mediated endocytosis and subsequently participates in the endosome pathway [33]. The role of CAV-1 in viral intracellular trafficking and the crosstalk between transcytosis is shown in Figure 2.

CAV-1 is also likely to participate in the sorting of viral components to different membrane surfaces. As early as in the 1980s, Rodriguez Boulan and colleagues found that the surface distribution of different viral envelope glycoproteins varies in infected epithelial monolayers. The glycoproteins of the influenza virus are mainly concentrated on the surface of the apical side, while the G protein of vesicular stomatitis virus (VSV-G) is at the basolateral side [74]. The influenza virus hemagglutinin (HA) is transported from the trans-Golgi network (TGN) to the apical cell surface by vesicles consisting of large homo-oligomers of CAV-1. VSV-G is transported from TGN to the basolateral surface by vesicles consisting of hetero-oligomers of CAV-1 and CAV-2. The transport route might be regulated by the phosphorylation of CAV-2. Anti-CAV-1 antibodies inhibit the apical transport of HA but do not affect the basolateral delivery of VSV-G, demonstrating that CAV-1 homo-oligomers make a difference in apical delivery but not basolateral trafficking [75] (pattern diagram is in Figure 2). Moreover, the Epstein–Barr virus (EBV) is transported from basolateral to apical surfaces by CAV-1-mediated entry and secreted into the saliva of EBV-infected individuals [76].

In this section, we first exemplify the viruses that use the caveosome route for transport and then discuss the existence of crosstalk between caveosome and endosome trafficking pathways. We generalize the roles played by caveolae and CAV-1 in intracellular trafficking during viral infection in Table 2, and further summarize the role of CAV-1 in regulating the transportation of viral components (see Figure 2). In general, clathrin-mediated endocytosis and subsequent transport are the most utilized intracellular pathway, but why the caveosome transcytosis still exists, how cargos choose their way, and what the specific role of CAV-1 and complex mechanisms is in those transport routes still remain issues that need to be addressed.

## 4. The Role of Caveolae/CAV-1 in Virus Replication

The viral replication mechanism remains the most extensive direction for antiviral study. Lipid rafts have been known to be involved in the replication via the recruitment of cellular molecules used for virus replication. CAV-1 is responsible for the direct localization of virions or viral proteins within the lipid rafts, raising the possible function of CAV-1 in the process of viral replication.

Dengue virus (DENV) is an arbovirus of the *Flaviviridae* family that causes dengue hemorrhagic fever and dengue shock syndrome with high mortality. Caveolar cholesterol-rich lipid raft microdomains play an important role in DENV polyprotein processing and replication, and DENV nonstructural protein 3 (NS3) could interact with CAV-1 [77]. It has been reported that HCV, another flavivirus, induces autophagy to enhance its own replication. HCV can specifically induce lipid rafts to locate autophagosomes, thereby mediating their RNA replication. Caveolin-1, the marker of lipid rafts, was also found in autophagosomes, revealing that caveolin-1 might play a role in HCV replication [78]. CAV-1 is also involved in the replication of influenza A H1N1, possibly based on the interaction between CAV-1 and matrix protein M2 [79].

The inhibitory effect of CAV-1 on viral replication has also been identified. For example, overexpression of CAV-1 significantly decreases the transcription level of HIV-1, and the decrease of CAV-1 expression leads to an enhancement of HIV-1 replication. As well known, HIV actually benefits from the activation of host inflammatory responses [104,105]. Further investigations suggested that CAV-1 suppresses the NF-κB p65 acetylation, a requirement for signaling activation [106,107]. Taking these studies together, CAV-1 inhibits HIV-1 replication by suppressing the activation of NF-κB-mediated inflammatory responses [80].

In this section, we associate caveolae/CAV-1 with virus replication (Table 2). Thus, CAV-1 should be taken into consideration in future replication-associated studies, especially when it is related to lipid rafts-mediated signal transduction.

## 5. The Role of Caveolae/CAV-1 in Virus Assembly and Egress

The role of CAV-1 in virus assembly and release cannot be strictly separated. Caveolae share the same composition with lipid rafts, which are associated with assembly and egress of various viruses, such as HIV-1, influenza, and measles virus [110]. Assembly of viruses requires centralizing of the different viral components at specific regions, and when the specific assembly site is on the plasma membrane, caveolae starts to be involved. MLV is one of the cases. Its assembly is on the plasma membrane and its entry depends on CAV-1. By pull-down and co-immunoprecipitation assay, MLV-Gag precursor is found to directly interact with CAV-1 via its CBD. Gag protein drives the assembly of MLV on the plasma membrane, while CAV-1 co-localizes with Gag on the plasma membrane, which affects the function of Gag protein on promoting assembly, thus impair the production of infectious virions in this process. Intriguingly, the CBD domain is highly conserved in most γ-retroviruses, indicating that a broad spectrum interaction may exist between Gag and CAV-1 [81]. However, as mentioned above, the existence of CBD only provides the possibility of interacting with CAV-1.

As for the egress, the naked virus usually releases from infected cells through the destruction of the plasma membrane. However, the enveloped virus obtains a host-derived lipid bilayer surrounding the nucleocapsid or core during budding [111]. During viral release, the host cell membrane usually bends, then gradually forms a neck, and, finally, fission occurs. In this process, CAV-1 is incorporated into the envelope of mature viral particles and regulates the efficiency of virus release.

DENV NS1 could be secreted as a soluble hexamer to the extracellular milieu. A study uncovered a direct interaction between DENV NS1 and CAV-1 by in situ proximity ligation. By inhibiting the classical protein secretion pathway and CAV-1, respectively, it was found that the DENV NS1 is released through the unconventional route most probably involved in CAV-1 [82]. One more recent study showed that DENV NS1 takes advantage of the intracellular cholesterol transporter chaperone of the CAV complex to be released from the mosquito cells without passing through the Golgi apparatus [83].

It has been identified that RSV infection causes CAV-1 to form a filamentous structure [84]. Subsequently, RSV proteins co-localize with CAV-1 in infected cells [85]. Further work has proven that CAV-1 is actively recruited to the RSV envelope, based on the interaction between CAV-1 and the viral G protein and actin cytoskeleton network. Moreover, RSV infection induces significant changes in the stoichiometry and biophysical properties of the caveolae-coated complex due to the increased cavin-1 protein (another caveolae structure protein) levels induced by infection [86].

Parainfluenza virus 5 (PIV5) is a negative-stranded RNA virus that is mainly used as a vaccine vector and is not related to diseases known to humans. CAV-1 is found to interact with PIV-5 matrix (M), nucleocapsid (NP), and hemagglutinin-neuraminidase (HN) proteins. One study pointed out that CAV-1 is essential for the interaction between M and HN, and reducing this interaction could affect the infection efficiency of PIV5. This study also provided evidence that the infectivity of PIV5 decreases when host cells are able to stably express dominant-negative CAV-1. Meanwhile, the lack of CAV-1 leads to defects in the production of viral particles. Moreover, CAV-1 appears to act as a bridge between viral components and lipid rafts by binding the viral proteins to locate them on the plasma membrane. This dynamic oligomerization of CAV-1 results in enhanced clustering of viral complexes on the plasma membrane and triggers the assembly and/or budding of the virus [87]. In addition, the CAV-1 binding motif on M is highly conservative, suggesting that CAV-1 might play a crucial role in a certain class of viruses. Another member of the *Paramyxovirus* family, the Newcastle disease virus (NDV), displays similar properties. Evidence showed that purified NDV particles contain lipid raft-associated CAV-1, suggesting that CAV-1 might be related to NDV budding or release [88].

Unlike previous cases, CAV-1 can also inhibit the virus egress. As earlier mentioned, TFV enters HepG2 via CAV-1-needed endocytosis, and another study explored that CAV-1 has an inhibitory effect on the egress of TFV virions [89]. They uncovered that TFV major capsid protein (MCP) starts to co-localize with caveolae at 60 hours post-infection, suggesting that caveolae are involved in the late stage of infection rather than the early or entry step. Knocking down CAV-1 increases the release of TFV virions significantly. Moreover, several caveolae structure proteins, including CAV-1, CAV-2, cavin-1, and cavin-2, are selected to locate on the capsid of TFV virions.

In this section, we summarized and found that only a small percentage of viruses utilize caveolae/CAV-1 involved assembly and egress, and these two steps are sometimes inseparable (Table 2). Most viruses, such as retroviruses, use a more conservative multi-subunit endosomal sorting complex required for transport (ESCRT) mechanism to accomplish budding and egress [112,113]. Collectively, existing research shows that caveolae and CAV-1 play different roles in the assembly and release of some viruses, even leading to the opposite effect. The unified model and the specific mechanism are required for further study.

## 6. The Role of Caveolae/CAV-1 in Virus Infection Related Signal Transduction

Over these years, CAV-1 has been considered a key factor associated with various signaling pathways [114]. Some of the signal transductions are related to viruses, showing effects of inhibiting or promoting infection, as well as the pathogenicity of viruses. We summarized the role of CAV-1-related signaling pathways in different virus infections. In this section, we first introduce the widely-studied HIV and HBV in detail, followed by the regulation of CAV-1 in other viral infection-related signaling.

### 6.1. The Role of Caveolae/CAV-1 in HIV Infection-Related Signal Transduction

It has been mentioned above that CAV-1 has a negative effect on HIV replication, mainly by inhibiting the expression of HIV protein. In addition, some other studies have revealed multiple ways in which CAV-1 participates in the signaling regulation of HIV infection. An early study has shown that co-expression of CAV-1 and HIV-1 is able to inhibit infection through the C-terminus hydrophobic transmembrane region of CAV-1, mainly by inhibiting the production of proviral DNA [90]. In human Langerhans cells (LCs), CAV-1 is co-localized with C-type lectin receptor langerin, which is capable of HIV capture and transportation of HIV into Birbeck granules (BGs), thus inhibiting HIV infection through the lysosomal degradation pathway [91].

Tat, a specific HIV protein, causes high expression of CAV-1 and thus restricts HIV-1 infection [92]. It can also induce Ras signal activation, thereby taking HIV across the blood–brain barrier (BBB). This process has been reported to depend on CAV-1 [93]. The authors also showed that Tat interacts with CAV-1 and can induce the downregulation of occludin and LRP-1 and upregulation of RAGE and RhoA during HIV infection, which may cause amyloid β-peptide (Aβ) deposition in the brain [94]. Except for Tat, Nef protein also has an interaction with CAV-1 during infection. Under normal circumstances, Nef can induce the impairment of high-density lipoprotein (HDL) mediate-cholesterol efflux, which is beneficial for HIV infection. CAV-1 is upregulated by Tat, and CAV-1 can eliminate this effect by interacting with Nef and thereby activating cholesterol efflux, resulting in the reduction of HIV infectivity [95]. Besides, Nef protein is further shown to affect the CAV-1 redistribution, perhaps by inducing the phosphorylation of CAV-1 [96].

Env protein of HIV can cause a sharp decrease in the number of CD4^+^ T cells via inducement of bystander apoptosis, thereby resulting in immune deficiency. This process relies on the viral proteins gp41 and Env-induced membrane hemifusion between envelope-expressing cells and target cells. CAV-1 can interact with gp41 and inhibits Env-induced membrane hemifusion, thus reducing HIV infection [97].

Among acquired immunodeficiency syndrome (AIDS) patients, long-term non-progressor (LTNP) is a special kind of being that can maintain high CD4^+^ T-cell counts and control HIV replication for years. Studies have suggested that single-amino-acid-exchange mutations in gene UBXN6 might be responsible for this phenomenon. It is also suggested that CAV-1 is involved in this process. When UBXN6 is knocked down, the expression level of CAV-1 is upregulated, and the replication of HIV-1 is inhibited. It is speculated that the interaction among CAV-1, valosin-containing protein (VCP), and UBXN6 has changes that affect HIV replication [98].

To sum up, HIV infection not only dramatically influences CAV-1 expression and its subcellular localization but also affects multiply signaling pathways, thereby preventing cholesterol from transferring to HDL. In addition, the viral protein produced by HIV-infected cells induces the upregulation of CAV-1, and the expression of CAV-1 reciprocally inhibits the replication of HIV. The role of caveolae/CAV-1 in HIV infection related signaling pathway is summarized in Table 2. Since CAV-1 plays multiply roles during HIV-1 infection and pathogenesis, CAV-1 can be used as a target for the treatment of HIV infection. For instance, as CAV-1 can restrict HIV-1 infection, CAV-1 mimetics or analogs may be used to treat HIV-1 infection.

### 6.2. The Role of Caveolae/CAV-1 in HBV Infection-Related Signal Transduction

As mentioned in the first section, the entry of HBV into HepaRG cells is caveolae-dependent. A recent study has shown that gene polymorphisms causing differences in CAV-1 expression are associated with HCC, implying that CAV-1 exerts a crucial role in HBV infection-induced diseases [100]. Studies have also shown that CAV-1 is highly expressed in HCC and hepatitis tissues and participates in vascular endothelial growth factor (VEGF)-mediated signaling pathways. The correlation among CAV-1, VEGF levels, and microvessel density has been observed, indicating that CAV-1 plays a role in tumor angiogenesis with VEGF [99].

In addition, the transcriptional level of CAV-1 is downregulated when transfecting HBV-encoded X protein (HBx) into a cultured HCC cell line in vitro. Further research has suggested that the transfection promotes methylation of CpG islands on the promoter region of the CAV-1 gene and thereby inhibits the expression of CAV-1 [101]. C-terminal truncated HBx (HBxΔC) is frequently detected in HCC and exerts a more potent oncogenic effect than full-length form. In vitro expression of HBxΔC results in the upregulation of CAV-1 transcriptional level, which could affect a series of downstream signaling pathways related to HCC tumorigenesis and metastasis [102]. The role CAV-1 plays in HBV infection and HBV-related HCC is also shown in Table 2.

### 6.3. The Role of Caveolae/CAV-1 in Other Viruses Infection-Related Signal Transduction

Herpes simplex virus type 1 (HSV-1), a member of the alphaherpes virus family, is one of the most prevalent and successful human pathogens. During HSV infection, the host CAV-1 is utilized by HSV to escape inducible nitric oxide synthase (iNOS)-mediated responses. It has been proven that CAV-1 co-localizes with iNOS in dendritic cells and inhibits nitric oxide production. CAV-1 deficiency results in a reduction of HSV infection [103]. Porcine reproductive and respiratory syndrome virus (PRRSV) is a small, enveloped positive-sense RNA virus belonging to the family *Arteriviridae*. Less CAV-1 expression and a higher level of HSP90 are observed in natural PRRSV-infected lungs. Unbalanced HSP90 and CAV-1 are found to be responsible for the increasing level of iNOS, thus inducing an increased reactive oxygen species (ROS) generation and higher endothelial dysfunction [104].

Infection of influenza A virus (IVA) is also associated with CAV-1-mediated signal transduction. During IVA infection, CAV-1 prevents the virus from infecting mouse embryo fibroblasts (MEFs). The study also showed that the absence of CAV-1 causes cell apoptosis in the late phase of viral multiplication via the upregulation of p53 and ROS levels. Strikingly, the elevated levels of ROS and apoptosis cause the increase of IVA virion yield. In other words, CAV-1 plays a role in limiting IVA infection [105].

Except for the viruses mentioned above, HPV16, rotavirus, Venezuelan, and western equine encephalitis viruses (VEEV and WEEV, respectively) are also related to CAV-1-associated signaling pathways. During HPV16 infection, viral protein E6 downregulates CAV-1 level by inactivation of p53, which also induces cells into malignant transformation process [106]. Rotavirus nonstructural protein (NSP) 4 was indicated to interact with CAV-1 through its residues 2–22 and 161–178 by using reverse yeast two-hybrid analyses and direct peptide binding assays, while the association of NSP4 and CAV-1 leads to NSP4 intracellular trafficking from ER to the cell surface, and exogenously added NSP4 may stimulate signaling molecules located in caveola microdomains [107,108]. A recent study indicated that VEEV and WEEV cross the BBB through CAV-1-mediated transcytosis, and interferons (IFN) and RhoA GTPase signaling pathways are involved in the viral uptake process [109].

In summary, CAV-1 mainly plays a role in activation or suppression in the following signal pathways in the context of virus infections: (1) Ras signaling pathway; (2) VEGF-mediated signaling pathways; (3) iNOS-mediated responses; (4) p53′s function; (5) IFN signaling pathways; (6) RhoA GTPase signaling pathways (summarized in Table 2). Together, CAV-1 not only participates in the viral life cycle but also relates to a number of diseases induced by virus infections, which make CAV-1 a potential conservative treatment site for clinical diseases. For example, a clinical trial that aims to study the relationship between CAV-1 and vascular dysfunction is ongoing. They hypothesize that genetic variation at the CAV-1 locus is a determinant of vascular phenotypes in hypertensive subjects [115]. Viral infections are sometimes associated with hypertension, and this clinical trial may further reveal the link between CAV-1 and diseases, providing data support for clinical applications. Besides, CAV-1 antagonists or inhibitors can be designed to block infections of PIV5, NDV, or used to cure diseases that are associated with elevated CAV-1. A drug named LTI-03, a CAV-1-scaffolding-protein-derived peptide, is also ongoing a clinical trial subject. This study will investigate the initial safety, tolerability, and PK profile of inhaled LTI-03 in healthy volunteers, and findings from this subject will direct the clinical development of LTI-03 for the treatment of diseases [116].

## 7. Conclusions and Perspective

Caveolae and CAV-1 have been identified to have a wide range of effects in all stages of viral infection. First and most important, caveolae-mediated endocytosis could serve as a mechanism for virus internalization (Table 1). Secondly, descriptive studies have provided rich insights into CAV-1-involved post-endocytic trafficking of viruses. Moreover, caveolae and caveolae/CAV-1 has been discovered to make differences in the process of replication, assembly, and egress, although the specific mechanisms in some cases have not been thoroughly elucidated yet. Finally, caveolae and CAV-1 function in a number of signaling pathways relevant to virus infection and viral pathogenesis (Table 2). We summarize a pattern diagram in different steps in which caveolae affects virus infection in Figure 1 (entry) and Figure 2 (trafficking, replication, assembly, and egress), and also show some easily understandable cases of caveolae/CAV-1-related signaling pathways in Figure 3.

As we are gaining a greater understanding of the regulation of caveolae/CAV-1 in the process of virus infections, there are numbers of exciting and substantial questions worth pursuing in the future:

(1) The most mysterious question on virus entry dependence is what we called “cherry-picking”. JEV enters BHK-21 cells via the clathrin-needed pathway, while it enters human neuroblastoma cells through the caveolae-dependent route. Why does one specific virus exhibit preferences on a particular endocytic pathway in different host cells? Can the compositions and receptors on the host cell membrane be one of the reasons?

(2) Viruses like TGEV and PEDV utilize both pathways to achieve entry to the host cells. Do clathrin- and caveolae-mediated endocytosis share any common bureaucracy? A in vivo complexity might be hidden to balance between these two endocytosis mechanisms;

(3) Virus-like HPV31 enters cells via the caveolar pathway and then shuttles towards endosomes, indicating the existence of a yet unknown machinery for the fusion of caveosomes and endosomes;

(4) The connection between host cell receptors and internalization mechanisms is worth explaining. In the *Coronaviridae* family, viruses such as HCoV-OC43 and HCoV-229E both employ caveolae for internalization, while SARS-CoV seems not to and prefers clathrin-mediated-endocytosis. A reasonable explanation may be because the cell receptors are different. SARS-CoV is reported to utilize ACE for entry, while HCoV-229E utilizes CD13 and HCoV-OC43 utilizes HLA class I molecule or sialic acids [117,118,119,120]. As mentioned in FMDV, only HS-mediated entry of FMDV is caveolae-associated, suggesting that the receptor is associated with the virus uptake mechanism.

(5) Are there any other “trafficking-associated-somes” or “intermediate-status-somes” other than caveosomes and endosomes? Does caveolae/CAV-1 extend any overlapping/redundancy functions in multiple virus trafficking pathways?

(6) How deep caveolae/CAV-1 is involved in regulating virus replication, the most concerning step, is uncertain, despite that studies have proven the co-localization of CAV-1 and viral proteins;

(7) Caveolae/CAV-1 can be incorporated into the virions of some viruses [86,88]. Whether this process occurs passively as viral budding or actively through interaction with viral proteins or host cell components is unknown;

(8) The assembly and budding sites of certain viruses are in the caveolae/CAV-1-riched lipid raft regions on host cell membranes, providing us a new field of vision concerning viral transmission control strategy from the CAV-1 lipid system;

(9) Caveolae/CAV-1, on many occasions, takes advantage of the same signaling upon distinct virus infections, indicating that small molecules can be designed to gain potential pan-antiviral compounds;

(10) Caveolae/CAV-1 is also associated with HCC, and causes brain Aβ deposition upon HIV infection, suggesting that caveolae/CAV-1 could serve as an important target for clinical treatment of many virus-derived diseases;

(11) Some viruses possess a conserved CAV-1 binding motif, providing us with a direction to explore CAV-1 as a broad-spectrum virus-binding protein.

We have witnessed a substantial increase in our understanding of how caveolae/CAV-1 regulates virus infection. Unified and holistic exploration is imperative and is starting now to provide new insights into caveolae/CAV-1 during virus infections, most interestingly, from the perspective of cell biology. Therefore, precise dissection of the underlying mechanisms of caveolae/CAV-1 is vital and will be an attractive and must-go direction for the discovery of new host-targeted antiviral strategies.

## Figures and Tables

**Figure 1 viruses-12-00487-f001:**
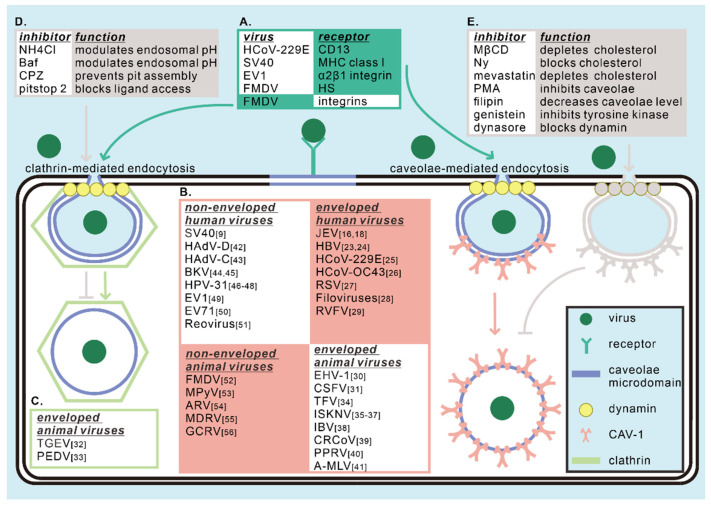
Caveolae/CAV-1 play critical roles in the entry of viruses to the host cells. (**A**) Firstly, the virus binds to a specific receptor in caveolae microdomain. The receptors involved in endocytosis are specifically described and listed. CD13, MHC class I, α2β1 integrin, and HS are involved in caveolae-mediated endocytosis, while integrins participate in clathrin-mediated endocytosis of foot-and-mouth disease virus (FMDV). (**B**) Subsequently, viruses internalize through the caveolae-mediated endocytic pathway are divided into four categories according to their structures and hosts. (**C**) It is noteworthy that transmissible gastroenteritis virus (TGEV) and porcine epidemic diarrhea virus (PEDV) were reported to complete endocytosis by both caveolae- and clathrin-mediated pathways. (**D**,**E**) Inhibitors of clathrin- and caveolae-mediated endocytosis and their related functions are described and listed.

**Figure 2 viruses-12-00487-f002:**
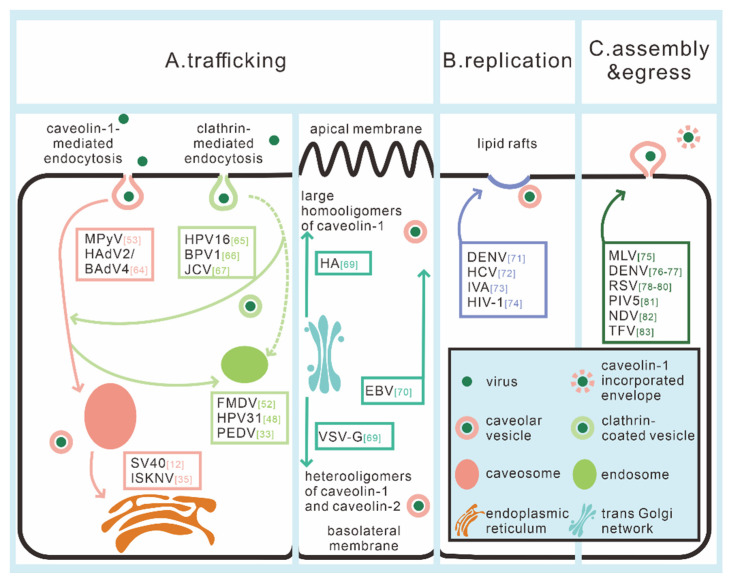
Multifunctional roles of caveolae/CAV-1 in the life cycle of corresponding viruses. (**A**) Trafficking: CAV-1 is involved in the complicated trafficking network of viruses. There is host cell type-, virus type-dependent crosstalk between caveosome trafficking and endosome trafficking, despite different endocytic pathways (caveolae- or clathrin-mediated) that viruses hijack. In addition to viral particles, CAV-1 is also involved in regulating the sorting and transport of viral components/proteins. (**B**) Replication: caveolae and CAV-1 facilitate the targeted location of virions, viral proteins, and host-related proteins to lipid rafts-associated compartments to complete viral replication. (**C**) Assembly and egress: caveolae and CAV-1 are involving in virus release and might incorporate into the enveloping process of some mature virus particles. The chromatic wireframes indicate the reported viruses involved in these processes. Numbers in square brackets indicate the relative references.

**Figure 3 viruses-12-00487-f003:**
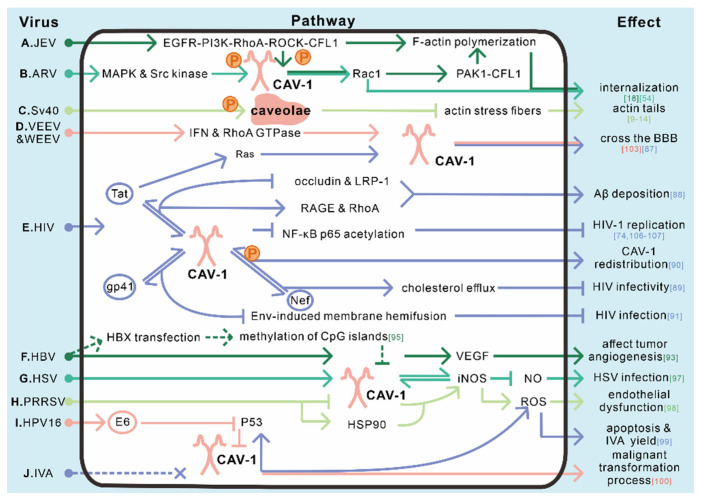
Summary of caveolae/CAV-1 in signal transduction during virus infection. A total of 16 caveolae and CAV-1-related signaling pathways from 11 viruses are shown in this figure. Different viruses and involved signaling pathways associated with caveolae and CAV-1 are represented in different colors to distinguish among complex networks. (**A**,**B**) The internalization of the Japanese encephalitis virus (JEV) into SK-N-SH cells activates the EGFR-PI3K-RhoA-ROCK-CFL1 signaling pathway and leads CAV-1 phosphorylation and furthers Rac1 activation, thereby initiating PAK1-CFL1-mediated actin polymerization and JEV internalization [18]. The entry of Avian reovirus (ARV) also takes advantage of the phosphorylation of CAV-1 and Rac1 activation, while by activating p38 MAPK and Src kinase [54]. (**C**) The transient depolymerization of the actin stress fibers causes the formation of actin tails, which depends on the local tyrosine phosphorylation of caveolae after SV40 infection [9,10,11,12,13,14]. (**D**) Venezuelan equine encephalitis virus (VEEV) and western equine encephalitis virus (WEEV) successfully cross the BBB with the help of IFN and RhoA GTPase signaling pathways via CAV-1-mediated transcytosis [109]. (**E**) CAV-1 plays a variety of roles during HIV-1 infection. Tat protein can activate the Ras signaling pathway, making HIV successfully cross the BBB with the dependency on CAV-1 [93]. The interaction between CAV-1 and Tat also results in less occludin and LRP-1 expression and more RAGE and RhoA production, thus making Aβ deposition in the brain [94]. In addition, CAV-1 restricts HIV infection in many ways. Firstly, CAV-1 inhibits HIV-1 replication by suppressing NF-κB p65 acetylation, which is indispensable for inflammatory response activation [80]. Secondly, CAV-1 could interact with HIV Nef protein to activate cholesterol efflux, thus leading to an impaired HIV infectivity [95]. Besides, Nef protein is also involved in inducing CAV-1 phosphorylation and further affecting its redistribution [96]. Lastly, CAV-1 reduces HIV infection by interacting with gp41, mainly through inhibition of Env-induced membrane hemifusion [97]. (**F**) CAV-1 plays a role in tumor angiogenesis through participates in VEGF-mediated signaling pathways during HBV infection [99]. In in vitro transfected HBx rather than an actual infection, CAV-1 expression was inhibited by promoting the methylation of CpG islands on its promoter region [101]. (**G**–**J**) iNOS, ROS, and p53 construct a complex network since they are involved in different CAV-1-related pathways during virus infections. In the case of HSV infection, CAV-1 plays a positive role in the infection via interacting with iNOS to inhibit the production of nitric oxide [103]. During PRRSV infection, less CAV-1 and higher levels of HSP90 are found to be responsible for the increasing levels of iNOS. Increased iNOS further results in higher ROS generation and endothelial dysfunction [104]. In CAV-1-lack cells, ROS and p53 are upregulated and cause an increase of IVA virion yield during IVA infection [105]. Moreover, the E6 protein of HPV16 could downregulate CAV-1 via p53-inactivating, which eventually causes malignant transformation [106]. Numbers in square brackets indicate the relative references.

**Table 2 viruses-12-00487-t002:** Summary of the roles of caveolae/CAV-1 in post-entry steps of virus infection.

Viral Life Cycle Steps	Viruses	Description	References
Intracellular trafficking	SV40	Go through a two-step transport route from plasma membrane caveolae to the ER through caveosomes	[12]
	ISKNV		[35]
	MPyV	Trafficking through the CAV-1-containing monopinocytic vesicles and the involvement of the cytoskeletal system	[53]
	HAdV2/BAdV4	Induces the formation of mega-caveosomes, and subsequent trafficking does not require endosomes	[70]
	HPV16	Entry via clathrin-mediated endocytosis, and shuttle from endosome pathway to caveolae route and then on to ER	[71]
	BPV1		[72]
	JCV		[73]
	FMDV	Enters cells via caveolar pathway and then shuttles towards endosomes	[52]
	HPV31		[48]
	PEDV	Enters cell through both clathrin- and caveolae-mediated way, and then participates in the endosome pathway	[33]
	VSV	VSV-G is transferred from TGN to the basolateral surface by vesicles consisting of hetero-oligomers of CAV-1 and CAV-2	[75]
	Influenza virus	HA is transported from TGN to the apical cell surface by vesicles consisting of large homo-oligomers of CAV-1	[75]
	EBV	Transports from basolateral to apical by CAV-mediated virus entry	[76]
Replication	DENV	Caveolar microdomains play a role in DENV polyprotein processing and replication	[77]
	HCV	HCV infection induces autophagy to enhance its own replication, and CAV-1 is found on autophagosomes	[78]
	IVA	Interaction between CAV-1 and M2 matrix protein affects H1N1 replication	[79]
	HIV-1	CAV-1 could inhibit the expression of HIV-1 protein	[80]
Assembly and egress	MLV	CAV-1 interacts with Gag, which could drive MLV assembly	[81]
	DENV NS1	CAV-1 assists DENV NS1 release from mosquito cells without passing through the Golgi apparatus	[82,83]
	RSV	Caveolar morphology changed; CAV-1 is incorporated into the envelope	[84,85,86]
	PIV5	CAV-1 triggers the assembly and/or budding of virus	[87]
	NDV	Purified NDV particles contain lipid raft-associated CAV-1	[88]
	TFV	CAV-1 has an inhibitory effect on the egress of TFV virions	[89]
Related signaling pathway	HIV	CAV-1 inhibits the production of proviral DNA	[90]
		CAV-1 inhibits HIV through lysosomal degradation pathway	[91]
		Tat induces CAV-1 upregulation	[92]
		HIV could cross the BBB owning to CAV-1-needed Tat-induced Ras signal activation	[93]
		Tat interacts with CAV-1, causing Aβ deposit in brain	[94]
		CAV-1 interacts with Nef, activating cholesterol efflux	[95]
		Nef affects the redistribution of CAV-1, thereby preventing cholesterol from transferring to HDL	[96]
		CAV-1 interacts with gp41 and inhibits Env-induced membrane hemifusion	[97]
		CAV-1 is involved in the mutation of UBXN6, which can affect HIV replication	[98]
	HBV	CAV-1 associates with HCC and participates in VEGF-mediated signaling	[99,100]
		HBx can downregulate CAV-1 transcription in HCC cell line	[101]
		HBxΔC can upregulate CAV-1 transcription in HCC cell line	[102]
	HSV-1	CAV-1 is utilized by HSV to escape iNOS-mediated responses	[103]
	PRRSV	CAV-1 and HSP90 induce an increased ROS generation and higher endothelial dysfunction	[104]
	IVA	The absence of CAV-1 leads to an upregulation of p53 and ROS level	[105]
	HPV16	Viral protein E6 downregulates CAV-1 level by inactivating p53	[106]
	Rotavirus	Rotavirus NSP 4 associates with CAV-1, leading NSP4 intracellular trafficking from the ER to the cell surface	[107,108]
	VEEV and WEEV	IFN and RhoA GTPase signaling pathways are involved in CAV-1-mediated transcytosis	[109]

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
