# Peer review of "Multifaceted Functions of Host Cell Caveolae/Caveolin-1 in Virus Infections"

_viruses, 2020, doi:10.3390/v12050487_

Round 1

Reviewer 1 Report

This is an excellent review manuscript that presents an ample compilation of the available literature on the role of caveolae and caveolin-1 during viral infections.  It’s timely due to the current pandemic and shall significantly contribute to the ever increasing demand for knowledge to assist us globally in the management of viral threats.  The outstanding features of the review reside in its ample compilation of available literature on the topic, well presented schematics in figures of inherently complex processes, and the posing at the end  of challenging questions for the field.

The main weaknesses are:

  1. The coalescence of the term Caveolae/Caveolin-1. This combined terminology is challenging and may be too encompassing, since caveolae and caveolin-1 may be separable for discussions.  For instance, there are Caveolae independent functions of caveolin-1.  In this respect specificity of the terms may be lost during the discussion and conclusions.  It’s my concern that readers of the manuscript may also feel uncomfortable with the coalescence of the terms.  Yet, in one way the amplitude of the cited literature may compensate for this perception.
  2. Section 2 on the entry of viruses via the caveolae-mediated pathway. The section can be significantly strengthened by including in Table 1 a column listing the experimental approaches used to conclude caveolae involvement in viral entry (EM, fractionations, Immunoprecipitations, pharmacological (filipin, B-Me-cyclodextrin, etc).  This has been successfully done in other literature reviews, i.e. GPCR (Chini, Parenti, 2009).  The manuscript  includes in limited instances this type of consideration:  MLV (Lines 321-322) and DENV (Line 331).  This will permit the authors (and readers) an assessment of the available experimental evidence claiming caveolae involvement in viral entry. This will nourish the text flow in this section beyond simply stating X virus also enters via caveolae, or is caveolae-mediated.
  3. Although posed as an important question (lines 513-515), a textual or tabular summary of surface receptors identified involved in viral entry must be included. This is an important aspect since it represents a significant void (Not Determined) in our body of knowledge, permits identification of potential therapeutic targets, and will enhance our understanding of entry behavior which are receptor-driven or mediated (via clathrin- or caveolae-).  Some virus-receptor pairs are dispersedly mentioned throughout the text:  HBV-HBsAg, HCov-229E-CD13, EV1-alpha2-beta1 integrin, and FMDV-HS/integrins. 
  4. Being a review on caveolae and caveolin-1, the authors quote when available the presence of caveolin binding domains (CBD) (Line 322, MLV); hence, giving value to their importance in the context of the review (Lines 538-539). Although the relative value of the linear CBD sequences has been questioned (Byrne DP, Dart C, Rigden DJ,  (2012), the authors must expand their consideration of the CBD in the context of the coronaviruses, particular SARS-Cov-1 and SARS-Cov-2.  Accordingly, must revise their statement in Lines 107 to 112 of having only two CoV related to CAV, in lights of the work by CAI Quan-Cai et al. (2003).  The in silico analysis presented in the latter study   identifies up to 32 putative caveolin-binding sites in SARS-CoV-1 proteins, which have been conserved in the SARS-CoV-2 (Yuan et al., Science 10.1126/science.abb7269 (2020)).  Certainly, the finding and conservation of putative CBD in molecules besides the spike protein may require their due consideration in other sections of the review. Yet, the current emergency demands readdressing and answering the most fundamental questions, such as why some coronaviruses enter in humans and animals via the CAV/cav1 paths, yet SARS Cov1/2 seemingly DO NOT (albeit an abundance of CBD) and  prefer clathrin-mediated-endocytosis?  As argued before, viral tracking strictly obeys receptor pharmacodynamics?  Inded, identifying the viral receptor repertoire is of crucial importance.
  5. Subsequent to an expose of a significant body of references, the review uses as a closing stratagem the posing of intriguing questions for the field. But, despite the vast body of information displayed, practical biomedically and clinically relevant conclusions or directions are missing.  In this respect, a section would be warranted (better than a one line statement, Lines 464-465), even if speculative, on what would be the expected benefits of manipulating the CAV/caveolin-1 system for viral therapeutics (indeed, as promised by the review).  For instance, can we use caveolin-1 mimetics or agents eliciting its upregulation to treat IVA (H1N1), HIV, MLV.  Conversely, shall we design caveolin-1 or CAV antagonists inhibitors or use siRNA or CRSPR technology to block infections by PIV5, NDV and TFV.  Are there promising therapeutic targets in the host of CBD expressed in the SARS-Cov-1/2 viruses?

The minor weaknesses are:

  1. Significant editing for grammar and spelling. As an example, the following table lists those found in the title and abstract section only.

Line

Original

Corrected/suggested

2

All-round…

Multifacetd… or Multifunctional …

15

…structure

…structural

15

…as its role

…for its role

22

…are caveolae/caveolin-1-required

…are caveolae/caveolin-1-dependent

23

…to the abnormalities

…to abnormalities

25

…and require holistically to be explored to provide potentially novel

… and must be explored holistically to provide potential novel

  1. Line 101. PH should be pH.
  2. Line 257. In Figure 1 check sentence may imply all viruses enter via caveolae 
  3. Lines 367 to 369. Abrupt insertion in sentence flow of clathrin at the end….  
  4. Table 2 is very clear delineating how caveolin-1 exerts anti-HIV actions. Yet, lines 411 to 413 throw in a mix of terms and concepts on HIV with no clear-cut statement of its clinical relevance or else
  5. Line 415. “… reversely inhibits…” seems contradictory, not clear.
  6. Substitute throughout the text the term “Signaling transduction” for “cell signaling” or “signal transduction”

Author Response

Reviewer #1

This is an excellent review manuscript that presents an ample compilation of the available literature on the role of caveolae and caveolin-1 during viral infections.  It’s timely due to the current pandemic and shall significantly contribute to the ever increasing demand for knowledge to assist us globally in the management of viral threats. The outstanding features of the review reside in its ample compilation of available literature on the topic, well presented schematics in figures of inherently complex processes, and the posing at the end of challenging questions for the field.

Response:

We thank the reviewer for positive feedback and excellent suggestions for strengthening our manuscript.

The main weaknesses are:

  1. The coalescence of the term Caveolae/Caveolin-1. This combined terminology is challenging and may be too encompassing since caveolae and caveolin-1 may be separable for discussions.  For instance, there are Caveolae independent functions of caveolin-1.  In this respect specificity of the terms may be lost during the discussion and conclusions.  It’s my concern that readers of the manuscript may also feel uncomfortable with the coalescence of the terms.  Yet, in one way the amplitude of the cited literature may compensate for this perception.

Response:

This is an excellent suggestion. We checked the manuscript very carefully and revised the text by specifying the function is caveoin-1 dependence or caveolae structures dependence according to specific references. It is worth noting that we try to inherit the original description accurately for every references used in our manuscript, but the various concepts are not clearly distinguished, for example, “caveolae-mediated endocytosis” (line 91,102) and “CAV-1-mediated endocytosis” (line 156) are both expressed by their respective authors. This problem can be solved perfectly only by completely standardizing the relevant statements in this field. Here we can only avoid the problem of unclear direction. Individual changes in the text include:

  1. In line 54,57,63,319,518,550,553 “caveolae/CAV-1” has been revised to “caveolae and CAV-1” based on the corresponding publications.
  2. In line 75, “Caveolae/CAV-1” has been revised to “Caveolae”
  3. In line 77, “Caveolae/CAV-1” has been revised to “Caveolae”
  4. In line 557, “Caveolae/CAV-1” has been revised to “Caveolae”

In addition, we also reviewed this issue carefully in all figures and legends, and updated figure 2 (now figure 3) and modified corresponding legends. Individual changes in the figures and legends include:

  1. In figure 3, for the accuracy of the description, all the concluding “caveolae cartoon” were replaced with “CAV-1 protein cartoon”, except that the SV40 pathway was indeed involved caveolae structure.
  2. In line 527, “caveolae” has been revised to “CAV-1”.
  3. In line 540, “caveolae” has been revised to “CAV-1”.
  4. In line 552, “caveolae” has been revised to “CAV-1”.

  1. Section 2 on the entry of viruses via the caveolae-mediated pathway. The section can be significantly strengthened by including in Table 1 a column listing the experimental approaches used to conclude caveolae involvement in viral entry (EM, fractionations, Immunoprecipitations, pharmacological (filipin, B-Me-cyclodextrin, etc). This has been successfully done in other literature reviews, i.e. GPCR (Chini, Parenti, 2009).  The manuscript includes in limited instances this type of consideration:  MLV (Lines 321-322) and DENV (Line 331).  This will permit the authors (and readers) an assessment of the available experimental evidence claiming caveolae involvement in viral entry. This will nourish the text flow in this section beyond simply stating X virus also enters via caveolae, or is caveolae-mediated.

Response:

We have now included a new column in Table1 demonstrating the experimental approaches used in the references as reviewer kindly suggested. Moreover, we summarized drugs (chemical inhibitors) and its mechanism which can be used to distinguish caveolae- and clathrin- mediated endocytosis in Figure 1. We further specified the methodology also in the text, including: CRCoV, PPRV, A-MLV (line 177-185), HAdV-C (line 196-198).

  1. Although posed as an important question (lines 513-515), a textual or tabular summary of surface receptors identified involved in viral entry must be included. This is an important aspect since it represents a significant void (Not Determined) in our body of knowledge, permits identification of potential therapeutic targets, and will enhance our understanding of entry behavior which are receptor-driven or mediated (via clathrin- or caveolae-).  Some virus-receptor pairs are dispersedly mentioned throughout the text:  HBV-HBsAg, HCov-229E-CD13, EV1-alpha2-beta1 integrin, and FMDV-HS/integrins. 

Response:

We have now included a new figure (Figure 1) summarizing known surface receptors of individual virus (Figure 1A), which might contribute to understand the receptor-driven of viral entry behaviors.

  1. Being a review on caveolae and caveolin-1, the authors quote when available the presence of caveolin binding domains (CBD) (Line 322, MLV); hence, giving value to their importance in the context of the review (Lines 538-539). Although the relative value of the linear CBD sequences has been questioned (Byrne DP, Dart C, Rigden DJ,  (2012), the authors must expand their consideration of the CBD in the context of the coronaviruses, particular SARS-Cov-1 and SARS-Cov-2. 

Accordingly, must revise their statement in Lines 107 to 112 of having only two CoV related to CAV, in lights of the work by CAI Quan-Cai et al. (2003). 

The in silico analysis presented in the latter study   identifies up to 32 putative caveolin-binding sites in SARS-CoV-1 proteins, which have been conserved in the SARS-CoV-2 (Yuan et al., Science 10.1126/science.abb7269 (2020)).  Certainly, the finding and conservation of putative CBD in molecules besides the spike protein may require their due consideration in other sections of the review. Yet, the current emergency demands readdressing and answering the most fundamental questions, such as why some coronaviruses enter in humans and animals via the CAV/cav1 paths, yet SARS Cov1/2 seemingly DO NOT (albeit an abundance of CBD) and prefer clathrin-mediated-endocytosis? 

As argued before, viral tracking strictly obeys receptor pharmacodynamics?  Indeed, identifying the viral receptor repertoire is of crucial importance.

Response:

This is an excellent suggestion, and thank the reviewer for relevant references. We have added explanation CBD in the text (line 117-119,357-363). In addition, we have added more of our understanding on the internalization mechanism of SARS-CoV in the text (line 114-127).

Moreover, we have read the paper reviewer suggested (A highly conserved cryptic epitope in the receptor-binding domains of SARS-CoV-2 and SARS-CoV, Science. 2020). This study mainly describes a conserved binding site of monoclonal antibody CR3022 which exists on the S protein of SARS-CoV and SARS-CoV-2, and there is no CAV-1 involved. Our tempting answer to the question why SARS-CoV-2 and SARS-CoV seems do not employ caveolae for internalization is possibly because of the different cell receptors. It has been reported that SARS-CoV reported to utilize ACEs for entry, HCoV-OC43 utilizes CD13, HCoV-229E utilizes HLA class I molecule or sialic acids. As mentioned in FMDV, only HS-mediated entry of FMDV is caveolae-associated, suggesting that receptor is associated with virus uptake mechanism. We add our understand of this issue in perspective (4) (Line 572-578).

  1. Subsequent to an expose of a significant body of references, the review uses as a closing stratagem the posing of intriguing questions for the field. But, despite the vast body of information displayed, practical biomedically and clinically relevant conclusions or directions are missing.  In this respect, a section would be warranted (better than a one line statement, Lines 464-465), even if speculative, on what would be the expected benefits of manipulating the CAV/caveolin-1 system for viral therapeutics (indeed, as promised by the review).  For instance, can we use caveolin-1 mimetics or agents eliciting its upregulation to treat IVA (H1N1), HIV, MLV.  Conversely, shall we design caveolin-1 or CAV antagonists inhibitors or use siRNA or CRSPR technology to block infections by PIV5, NDV and TFV.  Are there promising therapeutic targets in the host of CBD expressed in the SARS-Cov-1/2 viruses?

Response:

We have added in the end of signal transduction section (Line 506-515) to discuss the possible CAV/caveolin-1 targeted therapeutics. We further have added the idea of using CAV-1 analogs to treat HIV infection in line 454-457.

For SARS-CoV-1/2, there is no definite experimental data to prove that CAV-1 indeed play a role in their infection, so we cannot draw any definite conclusion that CAV-1 is useful for SARS-CoV-1/2 treatment at the moment.

The minor weaknesses are:

  1. Significant editing for grammar and spelling. As an example, the following table lists those found in the title and abstract section only.

Line

Original

Corrected/suggested

2

All-round…

Multifacetd… or Multifunctional …

15

…structure

…structural

15

…as its role

…for its role

22

…are caveolae/caveolin-1-required

…are caveolae/caveolin-1-dependent

23

…to the abnormalities

…to abnormalities

25

…and require holistically to be explored to provide potentially novel

… and must be explored holistically to provide potential novel

Response: We thank the reviewer for detailed checking on the title and abstract section, which we have modified accordingly. We have asked two native English speakers who are also scientists to carefully check the language of the manuscript, and modified several places for spelling errors, grammatical mistakes and confusions. We marked in the revised version of the manuscript with the "Track Changes" function.

  1. Line 101. PH should be pH.

Response: We have changed ‘PH’ to ‘pH’ as reviewer suggested (line 102).

  1. Line 257. In Figure 1 check sentence may imply all viruses enter via caveolae 

Response: We have made a separate figure (new Figure 1) to specify all viruses which enter host cells via caveolae, and implied all related viruses in the figure legends (line 249-257).

  1. Lines 367 to 369. Abrupt insertion in sentence flow of clathrin at the end….

Response: We have deleted the clathrin- part (line 406-407).

  1. Table 2 is very clear delineating how caveolin-1 exerts anti-HIV actions. Yet, lines 411 to 413 throw in a mix of terms and concepts on HIV with no clear-cut statement of its clinical relevance or else

Response: We have further added the idea of using CAV-1 analogs to treat HIV infection in line 454-457.

  1. Line 415. “… reversely inhibits…” seems contradictory, not clear.

Response: We have modified the sentence to ‘the expression of CAV-1 reciprocally inhibits the replication of HIV’. Line 450-454 is trying to demonstrate that viral protein of HIV increases CAV-1 expression, and expression of CAV-1 reciprocally inhibits the replication of HIV.

  1. Substitute throughout the text the term “Signaling transduction” for “cell signaling” or “signal transduction”

Response: We have changed the term “Signaling transduction” to “signal transduction” in the text, which are in line 350, 413, 485, respectively.

Reviewer 2 Report

This is a reasonably informative review that offer the opportunity to summarize the state of the art about our knowledge on the role of Caveolae/Caveolin-1 on viral infection. However, the manuscript requires essential revisions to be acceptable for a publication.

To improve the clarity of the review a new figure on the entry mechanism of virus via caveolae should be added. Figure should report the inhibitors (and the steps affected) that can be used to investigate the effect of caveolae on viral entry.

Major

Change the title of the paragraphs 2.1., 2.2., 2.3. and 2.4. (as well as the in the “classification” column of the figure 1 and figure legends) to improve clarity. In my opinion, the sentence “… entry of human susceptible enveloped viruses” is not correct.

Lines 37-38: Please revised the sentence with more details to improve clarity.

Table 1: there is a typo about the Filoviridae name

Lines 120-123 (and table 1):

  • use the official short name for Ebola virus and Marburg virus (see Kuhn JH et al., Nat Rev Microbiol. 2019 May;17(5):261-263. doi: 10.1038/s41579-019-0187-4).
  • What is EBO-Z? This is not an official nomenclature, it should be clarified.
  • Endocytosis is not a relevant pathway for filovirus entry. It can be shown with pseudoviruses but not with real filoviruses that require macropynocytosis (see some recent reviews like: Hoenen T et al., Nat Rev Microbiol. 2019 Oct;17(10):593-606. doi: 10.1038/s41579-019-0233-2; or Salata C et al., Viruses. 2019 Mar 19;11(3):274. doi: 10.3390/v11030274).

Line 122: the Bunyaviridae family is no longer used in the viral taxonomy. See Abudurexiti A et al., Arch Virol. 2019 Jul;164(7):1949-1965. doi: 10.1007/s00705-019-04253-6; or the International Committee on Taxonomy of Viruses web site.

Lines 135-137: Please revised the sentence including an interesting detail reported in the reference 31 “immunofluorescent confocal microscopy analysis suggested that CAV1 was temporally colocalized with CSFV E2 throughout the course of the infection”. In addition, authors should report that “CAV1-mediated endocytosis is advantageous for productive CSFV Shimen infection in macrophages” but not essential, because “RNA silencing of CAV1 did not prevent viral replication, which may indicate that CSFV can also enter macrophages by other mechanisms”,

It is important to understand when caveolin-mediated entry is the main/only route for viral entry and when it is an additional via, as reported in the same paragraph about the following Transmissible gastroenteritis virus and sometimes for other viruses. This information is very important and required for all the viruses reported in the review article.

Lines 257-258: Please revised the sentence to improve clarity. It is misleading.

Lines 300-302: Please revised the sentence to improve clarity.

Lines 303-304: It is not correct include influenza virus in the paragraph about replication (viral genome replication/transcription). The interaction between CAV-1 and M2 can affect the entry stage or the assembly/exit, not the genome replication.

Lines 323-324: Please revised the sentence to improve clarity. Why does CAV-1 impair the production of infectious virions?

Lines 340-342: Please revised the sentence to improve clarity.

Lines 411-413: Please revised the sentence to improve clarity.

Minor

Line 101: change PH with pH

Line 144: suspect cells probably is susceptible cells

Line 150: the name of the viral family have to writhe in italics (as in the line 147).

Line 182: papillomaviruses 31 is papillomavirus 31, in addition COS cells (monkey) are not “natural host cells” for the human papillomavirus 31.

Line 300: “Flaviviridae virus” should be changed in “flavivirus”

Author Response

This is a reasonably informative review that offer the opportunity to summarize the state of the art about our knowledge on the role of Caveolae/Caveolin-1 on viral infection. However, the manuscript requires essential revisions to be acceptable for a publication.

Response:

We thank the reviewer for thorough reading of our manuscript, and for his/her valuable suggestions aiming to improve the quality of our manuscript. We have taken these comments very seriously, and have modified accordingly.

To improve the clarity of the review a new figure on the entry mechanism of virus via caveolae should be added. Figure should report the inhibitors (and the steps affected) that can be used to investigate the effect of caveolae on viral entry.

Response:

Manuscript and references were carefully checked to standardize the description of inhibitors or drug, and we added and modified the related description. In line 176, we added “Mevastatin”. Moreover, we have made one more figure (new Figure 1) to specify all viruses which enter host cells via caveolae, and imply all related inhibitors (Figure 1D,E), receptors (Figure 1A) and experimental approaches (new column in Table 1) mentioned in this review.

Major

Change the title of the paragraphs 2.1., 2.2., 2.3. and 2.4. (as well as the in the “classification” column of the figure 1 and figure legends) to improve clarity. In my opinion, the sentence “… entry of human susceptible enveloped viruses” is not correct.

Response:

We have rephrased the title of paragraphs 2.1., 2.2., 2.3. and 2.4. to ‘… entry of enveloped human susceptible viruses’ as reviewer kindly suggested.

Lines 37-38: Please revised the sentence with more details to improve clarity.

Response:

We have specified the hijacked host mechanisms with several examples, which include ribosomes, cytoskeleton and clathrin (line 38-42).

Table 1: there is a typo about the Filoviridae name

Response:

We thank the reviewer for kind reminding, and we have modified the typo in Table 1

Lines 120-123 (and table 1):

  • use the official short name for Ebola virus and Marburg virus (see Kuhn JH et al., Nat Rev Microbiol. 2019 May;17(5):261-263. doi: 10.1038/s41579-019-0187-4).

Response:

We have changed to the official short name for Ebola virus (EBOV) and Marburg virus (MBGV) as reviewer kindly suggested (Line 135 and Table 1).

What is EBO-Z? This is not an official nomenclature, it should be clarified.

Response:

We have changed “EBO-Z” into Zaire EBOV (line 136).

Endocytosis is not a relevant pathway for filovirus entry. It can be shown with pseudoviruses but not with real filoviruses that require macropynocytosis (see some recent reviews like: Hoenen T et al., Nat Rev Microbiol. 2019 Oct;17(10):593-606. doi: 10.1038/s41579-019-0233-2; or Salata C et al., Viruses. 2019 Mar 19;11(3):274. doi: 10.3390/v11030274).

Response:

We have changed in the text demonstrating that the relevant endocytosis pathway was investigated by pseudoviruses, but not real live viruses (line 137-138).

Line 122: the Bunyaviridae family is no longer used in the viral taxonomy. See Abudurexiti A et al., Arch Virol. 2019 Jul;164(7):1949-1965. doi: 10.1007/s00705-019-04253-6; or the International Committee on Taxonomy of Viruses web site.

Response:

We thank the reviewer for kind reminding, and we have changed the Bunyaviridae family to ‘Phenuiviridae’ based on the reference reviewer suggested (line 139 and table 1).

Lines 135-137: Please revised the sentence including an interesting detail reported in the reference 31 “immunofluorescent confocal microscopy analysis suggested that CAV1 was temporally colocalized with CSFV E2 throughout the course of the infection”. In addition, authors should report that “CAV1-mediated endocytosis is advantageous for productive CSFV Shimen infection in macrophages” but not essential, because “RNA silencing of CAV1 did not prevent viral replication, which may indicate that CSFV can also enter macrophages by other mechanisms”

Response:

We have revised the sentence concerning reference 31 (line 154-158) to make it clear understood. We have added the two points as reviewer kindly suggested in the text.

It is important to understand when caveolin-mediated entry is the main/only route for viral entry and when it is an additional via, as reported in the same paragraph about the following Transmissible gastroenteritis virus and sometimes for other viruses. This information is very important and required for all the viruses reported in the review article.

Response:

This is an urgent problem that needs to be solved. In order to show the current situation more clearly, we modified the description of viruses that involve two endocytosis in the manuscript:

  1. In lines 161-163, TGEV
  2. In line 165, “suspect cells” has been revised to “into not just Vero cells and IPEC-J2 cells”

We also clarified this issue in the summary of the entry section:

  1. In line 239-242, “Besides, inhibition of clathrin- and caveolae-mediate endocytosis both affect infection in some cases, indicating the possibility of coexistence between the two pathways. Unfortunately, we do not find a pattern of which endocytosis the virus chooses, at least in terms of host cells.” has been added.

In the Conclusions and perspective section question (1) and (2) (line 562-569), we continue to see this as a challenge to be addressed in the future. And hopefully we will understand what kind of circumstances or conditions that allow the virus to choose between multiple endocytic pathways.

Lines 257-258: Please revised the sentence to improve clarity. It is misleading.

Response:

Since the original Figure 1 (now Figure 2) and legend oversimplified the description of the entry section, we added a new figure to elaborate the details for entry part and deleted this sentence in Figure 2 legend.

Lines 300-302: Please revised the sentence to improve clarity.

Response:

We have revised the sentence as “It has been reported that HCV, another flavivirus, induces autophagy to enhance its own replication. HCV can specifically induce lipid rafts to locate autophagosomes, thereby mediating their RNA replication. Caveolin-1, the marker of lipid rafts, was also found in autophagosomes, revealing that caveolin-1 might play a role in HCV replication” to make it clear to understood (in lines 335-339).

Lines 303-304: It is not correct include influenza virus in the paragraph about replication (viral genome replication/transcription). The interaction between CAV-1 and M2 can affect the entry stage or the assembly/exit, not the genome replication.

Response:

We checked this reference again and found that CAV-1 indeed affect the replication step of infection. This study points out in discussion section that “Taken together our results demonstrate, that Cav-1 exerts an influence on influenza A virus replication and data imply that the binding of Cav-1 to the matrix protein M2 is involved. However, which function or pathway in MDCK cells actually is triggered via Cav-1 interaction with M2, remains to be determined.”

What’s more, when other reviews cited this article, they also mentioned the impact is on replication steps, see review article (Int J Mol Sci. 2017 Dec 7;18(12):2649. doi: 10.3390/ijms18122649. Influenza A Virus M2 Protein: Roles From Ingress to Egress), they said “The M2/cav-1 interaction modulates IAV replication but the exact molecular mechanisms are not yet known”. In another review (Cells. 2019 Jun 29;8(7):654.  doi: 10.3390/cells8070654. Viroporins in the Influenza Virus), they also said “The M2/Cav-1 interaction has been found to be involved in modulating IAV replication, but it remains to be determined whether this interaction also affects the surface localization of M2 and other viral proteins”.

Therefore, our hypothesis is that maybe except the well-known function of M2 in early and late step of infection, it also can affect replication with the interaction with CAV-1.

Lines 323-324: Please revised the sentence to improve clarity. Why does CAV-1 impair the production of infectious virions?

Response:

We have revised the sentence as ‘Gag protein drives the assembly of MLV on plasma membrane, while CAV-1 co-localizes with Gag on plasma membrane, which affects the function of Gag protein on promoting assembly, thus impair the production of infectious virions in this process’ to make it clear to understand (line 358-361).

Lines 340-342: Please revised the sentence to improve clarity.

Response:

We have revised the sentence as ‘RSV infection induces significant changes in the stoichiometry and biophysical properties of caveolae coated complex due to the increased cavin-1 protein (another caveolae structure protein) level induced by infection’ to make it clear to understand (line 379-381).

Lines 411-413: Please revised the sentence to improve clarity.

Response:

We have revised the sentence as ‘To sum up, HIV infection not only dramatically influences CAV-1 expression and its subcellular localization, but also affects multiply signaling pathways, thereby preventing cholesterol from transferring to HDL.’ to make it clear to understand (line 450-452).

Minor

Line 101: change PH with pH

Response: We have changed ‘PH’ to ‘pH’ as reviewer suggested (line 102).

Line 144: suspect cells probably is susceptible cells

Response: We specific the susceptible cells to ‘Vero cells and IPEC-J2 cells ’.(line 165)

Line 150: the name of the viral family have to writhe in italics (as in the line 147).

Response: We have changed the format of ‘Iridoviridae family’ in italics, and checked throughout the text to make sure all the viral family have been written in italics. (line 168)

Line 182: papillomaviruses 31 is papillomavirus 31, in addition COS cells (monkey) are not “natural host cells” for the human papillomavirus 31.

Response: We thank the reviewer for kind reminding, and we have changed ‘papillomaviruses 31’ to ‘papillomavirus 31’ and remove the phrase of ‘natural host cells’ for COS cells. (line 207)

Line 300: “Flaviviridae virus” should be changed in “flavivirus”

Response: We have changed ‘Flaviviridae virus’ to ‘flavivirus’ as reviewer suggested. (line 336)

Round 2

Reviewer 2 Report

The revised version of the manscript is improved.

Minor corrections:

Please, change Marburg virus (MBGV) with the correct version Marburg virus (MARV)

Title of paragraphs 2.1., 2.2., 2.3. 2.4., and in figure 1: remove the word “susceptible”

Paragraph 2.4: whta is the correct version? domain-negative mutants OR dominant-negative mutants?

Author Response

Minor corrections:

Please, change Marburg virus (MBGV) with the correct version Marburg virus (MARV)

Response: We have changed to ‘Marburg virus (MARV)’ as reviewer kindly suggested (line 140).

Title of paragraphs 2.1., 2.2., 2.3. 2.4., and in figure 1: remove the word “susceptible”

Response: We have removed ‘susceptible’ from both text and figure 1.

Paragraph 2.4: what is the correct version? domain-negative mutants OR dominant-negative mutants?

Response: We have changed to ‘dominant-negative’ in the text (line 86).